# Orthodox Soil Science versus Alternative Philosophies: A Clash of Cultures in a Modern Context

**Robert E White [1],* and Martin Andrew [2]**

[1]  Faculty of Veterinary and Agricultural Sciences, The University of Melbourne, Parkville, Victoria 3010, Australia

[2]  Martin Andrew Solutions, Toorak Gardens, Adelaide, SA 5065, Australia; mcandrew@myaccess.com.au

*  Correspondence: robertew@unimelb.edu.au

**Abstract:** In Australia, orthodox soil scientists dealing with land management and alternative practitioners who promote 'regenerative agriculture' have not been communicating and engaging effectively with each other. Over many years, scientists in the Commonwealth Scientific and Industrial Research Organization (CSIRO), state departments and universities have made significant achievements in mapping soil distribution, describing soil behaviour and identifying key soil properties and processes that are fundamental to healthy soil function. However, many alternative practitioners are dismissive of these achievements and highly critical of orthodox soil science. Yet many of the tools of soil science are essential to conduct evidence-based research towards elucidating how and why the exceptional results claimed by some alternative practitioners are achieved. We stress the importance of effective engagement and communication among all parties to resolve this 'clash of cultures'.

**Keywords:** orthodox soil science; alternative practices; corrective strategies

## 1. Introduction

In 1959, the English author C. P. Snow published 'The Two Cultures and the Scientific Revolution' [1], decrying the fact that Western society was divided into two cultures—science and the humanities—and that neither understood what the other group was talking about. The division today between orthodox soil scientists and 'alternative' practitioners is another example of two cultures neither understanding each other nor communicating effectively. Both groups have the objective of improving soil and land management, but in many cases the alternative practitioners cannot provide scientific evidence as to why their practices are successful, nor do they accept that orthodox soil science has anything to offer in solving problems. In this article, we give Australian examples of this lack of mutual understanding. We explore reasons why this is occurring, the consequences of this disjunction and possible remedies for this unproductive clash of cultures.

## 2. Examples of the Disregard for Orthodox Soil Science in Australia

In 2006, a horse breeder and farmer named Peter Andrews published the book 'Back from the Brink' [2] in which he described his methods of regenerating the landscape through natural sequence farming. Another example is that of Charles Massy, a farmer in Southern New South Wales who in 2017 published the book 'Call of the Reed Warbler' [3], which deplores modern industrial farming—a product of what he calls the 'Mechanical Mind'—and promotes his practice of regenerative farming.

Both practitioners claim success on their own properties (which support grazing enterprises) through restoration of native grasses, increased biodiversity, enhanced storage of soil water, improved nutrient cycling and soil fertility, and minimizing erosion. But the explanations of these

outcomes are often dubious, taking no account of site and landscape-specific soil characteristics, and hence confounding the effective translation of the recommended practices to other situations. This deficiency highlights the fact that scientifically tested methods of grazing management [4] are not acknowledged. Nor is there recognition of the accumulated scientific knowledge on soil organic matter transformations [5], the many functions of the soil microbial community [6], the importance of nutrient balances [7] and methods of water management [8]. Nor is there any transparent cost-benefit analysis of the advocated practices.

Both authors are highly critical of established science. For example, Andrews [2] (p. 7) writes that 'we need the scientific community to accept that the approach it has adopted to Australia's landscape problems so far have been wrong' and that 'we certainly have to abandon the idea that scientists can provide a solution to our landscape's problems' [2] (p. 13). Massy [3] (p. 179) asks a rhetorical question about 'established sources of knowledge—department of agriculture people and the Commonwealth Scientific and Industrial Research Organization (CSIRO)—no they're just so far behind'. He also criticizes the 'unsustainable practices ... of modern industrial agriculture' [3] (p. 188), which have left a 'gutted, poisoned dysfunctional wasteland' [3] (p. 257). However, these opinions do not present a true picture of the contributions made by orthodox soil science, past and present, to solving problems associated with natural ecosystems and agroecosystems, as we illustrate in Section 3.

Notwithstanding the above, both books have been extensively praised and their ideas promoted. For example, the Australian Broadcasting Corporation (ABC) has featured the achievements on Peter Andrews' home farm in Central NSW on two 'Australian Story' TV programs and another such program on the activities of the Mulloon Institute in NSW, which promotes his practices of landscape management. Charles Massy's book received a very favourable review by Geordie Williamson, the chief literary critic of *The Weekend Australian Review* [9] (p. 16). Williamson was highly critical of the pioneer settlers who 'arrived with our Enlightenment certitude, our mechanistically minded arrogance, and royally screwed up everything'. Although there is no doubt that many of the approaches of European settlers to the manipulation of the Australian landscape for agriculture were misguided, land management and soil science thinking have evolved enormously since early times.

## 3. Promotion of Practices Alternative to Orthodox Soil Science

In a similar critical vein to Andrews and Massy, the organization Healthy Soils Australia (HSA) states on its website [10] that 'in the view of traditional soil scientists, soil is merely a porous medium for holding water and keeping plants upright. The role of microorganisms is seen as a little understood, and certainly not as important as climate, geography and soil chemistry'. Healthy Soils Australia promotes itself as being 'at the forefront of innovative thinking about how to apply our understanding of how soil works to: promote hydrological cooling, draw down atmospheric $CO_2$, and enhance agricultural productivity' [10].

Healthy Soils Australia is closely linked to Soils for Life [11], another not-for-profit organization. Soils for Life was founded by retired Major-General and former Australian Governor-General Michael Jeffery, who in 2013 was appointed by the Australian Prime Minister as the National Advocate for Soil Health. Soils for Life promotes the case studies of farmers who are practising 'regenerative agriculture' involving a change in their management practices to improve their soils and make their farming enterprises more sustainable and profitable. However, in the reporting of these case studies there appears to be little evidence of scientific measurements of changes in soil properties, nor of any detailed economic analysis of business improvement.

In a recent report to the Australian Prime Minister entitled 'Restore the soil: prosper the nation' [11], the National Advocate states that 'the reasons why the innovative methods developed by Soils for Life and other farmers are working so well are generally not well understood by science. More research is needed into the microbiological processes in the plant and soil biomes thought to be responsible for the success of various regenerative farming practices'. Few soil scientists would argue with the latter statement as an objective, and the report recommends 'collaboration between scientists and successful

farmers to build knowledge, collate the evidence to support successes and improvements and promote the wider use of regenerative farming techniques'. We agree with this call for collaboration. Regrettably, the report does not acknowledge the wealth of information derived from decades of Australian scientific observations and experiments that have examined interactions among soils, crops, pastures and animals. This omission helps to perpetuate the view that the science underpinning landscape management has not progressed since the early 1800s. The reality is entirely different. A variety of systems have been studied using a wide range of biophysical and chemical methods and modelling. Examples of this research extend from the earliest studies of Marston and co-workers [12], who unravelled the role of cobalt and copper in 'bush sickness' of ruminants, to Prescott [13], who studied soil formation and mapped soil distribution, to Norrish [14] elucidating the behaviour of soil clays, to Rovira [15] studying soil-borne diseases, to Lee and Foster [16] studying the interaction of soil fauna and soil structure, and to Baldock and co-workers [17] studying the dynamics of soil organic carbon and its measurement.

Notwithstanding the success of these studies, it is also true that the implementation of some research results has had unintended consequences for the sustainability of agricultural systems. For example, the widespread adoption in Southern Australia of pastures based on sub clover (*Trifolium subterraneum*) has led to accelerated acidification in soils that were poorly buffered [18]. However, field research has evolved from relatively simple plot-based experiments to large-scale ecosystems studies. This progression demonstrates that soil scientists have not been static in their thinking. The science has evolved and continues to do so. Nonetheless, in the broad context of the physical, chemical and biological interactions that determine healthy soil function, the Advocate's report pays scant attention to the research capabilities and achievements of CSIRO, state departments of agriculture and the universities, and their potential for collaboration with the proponents of alternative systems.

Thus, the disjunction between the 'two cultures'—that of orthodox soil science based on Nature's biophysical laws and rigorous measurement, and that of various alternative practices that must be accepted at face value—is clear. This situation appears to be in stark contrast to the advocacy for soil health research and extension set out in the strategy of the Soil Health Institute of the USA [19].

## 4. The Soil Health Institute Strategy

In 2013, agricultural industry leaders, farmers, ranchers, government agency leaders and non-governmental organizations gathered to examine and discuss the current state of soil health in the USA. From these discussions, the need was identified for a collaborative-oriented organization to provide 'accurate science-based information . . . .and coordinate change leadership'. The result was the formation of the Soil Health Institute [19], an independent not-for-profit organization focused on fundamental and applied research, but with a remit to bring knowledge from the research laboratory to the farm field. Knowledge generation and adoption are to be achieved through measurements and standards, economic analysis, communication, education and policy initiatives. The means of achieving these broad objectives are set out in a detailed and comprehensive strategic plan involving scientists and farmers [19], which could be a template for the kind of strategic approach adumbrated in the report 'Restore the soil: prosper the nation'.

The Soil Health Institute has a former senior scientist from the United States Department of Agriculture–Agricultural Research Service as chief executive officer and a professor of soil science at Texas A and M University as its incoming chief scientific officer. In contrast, Soils for Life [11], led by retired Major-General Jeffery, has no high-profile soil scientist on its board nor on its operating team; similarly for Healthy Soils Australia [10].

Initiatives like the Soil Health Institute are not foreign to Australia. The Sustainable Grazing Systems program (SGS), led by Meat and Livestock Australia in conjunction with other funding partners, was a 6-year program of collaboration between farmers and researchers with many insights gained from working on farmers' properties [4]. (The present authors were research participants in SGS).

## 5. A Way forward for Australian Soil Science

In the conferences on Global Soil Security held so far, the focus has been on the five 'dimensions' of soil security: Capability, condition, capital, codification and connectivity [20]. The last-mentioned of these could apply to our contribution here. However, the question should be asked: To what extent are these concepts meaningful to practical land managers and how can they assist such managers to 'secure' their soils and improve their management and the profitability of their enterprises? In addressing such questions, and with a view to improving mutual understanding between soil scientists and alternative practitioners, we would recommend the following four concepts: Communication, engagement, confidence and credibility.

### 5.1. Communication and Credibility

First, soil scientists need to communicate in a language that farmers and their champions in the media understand. This does not mean dumbing down the science, but it does mean they must challenge the unorthodox views of the alternative practitioners and seek explanations using language that an intelligent layperson can understand.

One example of an unorthodox view is reference [3] (p. 201) that cites a case study for which it is stated 'despite no superphosphate for over 35 years, phosphorus and other trace element and mineral levels have risen substantially soil pH having jumped from high acidity levels to nearly neutral'. Bearing in mind the law of conservation of mass, how can these increases have occurred in a production system unless there were substantial inputs of materials from off-site during the 35 years? Could the reported changes be explained by large inputs of manure from rotational grazing by 3000 sheep that would be expected not only to increase soil phosphate, but also to raise the soil pH?

Another example is the claim in reference [3] (p. 140) that a 1% increase in soil carbon (C) allows 144,000 L of extra water to be stored per ha to 0.3 m depth. This corresponds to an increase in the amount of stored water to 0.3 m depth of 14.4 mm. No information is given about the soil type for this claim, but from the scientific literature [21] we know that the effect of organic matter in increasing soil water storage is relatively more important in sandy than clay soils. Thus, the results that Morlat and Chaussod [22] obtained from a 28-year experiment in France with compost and manure on a soil comprising 86% sand are instructive. Compared to the control soil, organic C approximately doubled from 0.63 to 1.21% for soil treated with 20 t/ha/y of fresh cow manure. However, the increase in available water in the top 0.3 m was only 7.5 mm and this took 28 years to achieve. This is a much smaller response than that claimed in reference [3], so the soil and treatment conditions under which the reference [3] results are claimed need to be explained.

### 5.2. Communication and Engagement

Communication is a two-way street. Effective communication means real engagement. Agricultural science and soil science in Australia have a strong tradition of advancing knowledge through collaboration among farmers and scientists. Farmers have frequently identified outliers of success and scientists, who are professionally curious, have become eager collaborators to understand what is going on and what insights can be disseminated more widely. One such example is the aforementioned national SGS program. In that program, farmers and researchers were able to exchange ideas and information in the planning and execution of field-based experiments focusing on the productivity, environmental impacts and economic viability of grazing systems in Southern Australia's high rainfall zone [4]. There are other examples where farmers and community groups (often referred to as citizen scientists) have worked closely with soil scientists (often for decades) on catchment-related projects. This collaboration has led to substantial contributions and joint publications in international journals, which have often dealt with practical soil management and methods to improve several forms of agriculture [23,24]. As such, it is disappointing that many alternative practitioners have been

dismissive of these kinds of joint achievements and highly critical of orthodox soil scientists who deal with land management.

Soil scientists have offered to engage with Soils for Life and Healthy Soils Australia to investigate and understand the underlying processes in their exemplar sites. These are rich opportunities for understanding and making progress. However, little has happened, and one might ask the question: Why do the alternative practitioners not want collaborative, objective enquiry into these sites? Could it be that the lessons being promulgated are not so general after all?

*5.3. Credibility and Confidence*

It is only by communication; that is, by asking and answering questions in mutually comprehensible terms, can we hope to progress our understanding of these complex biological systems, and for the two cultural groups to have confidence in each other's results. As with enthusiastic farmers who have tried new methods and had success, soil scientists too are excited by unusual or even extreme behaviour in the soil-plant-animal systems being studied. However, they need to have full information about the conditions under which unusual results are obtained in a practical situation. From a study of these special cases, they can provide feedback to the practitioners that should enable further improvements to be made and the conditions specified whereby such practices can be translated with confidence to other soils and land use systems. This could provide substantive benefits to Australian farmers, international science and the Australian economy.

## 6. Conclusions

Australian soil scientists have enjoyed many successes in elucidating the properties of soils and how these properties, interacting with plants and the environment, influence soil processes critical to healthy soil function. They have also been at the forefront in identifying the distribution of soil properties at high spatial resolutions that give good predictive capabilities for productive and conservative soil management. However, there is still much to be done to understand the full integration of biology, chemistry and physics around soil C—the relationship between soil aggregation, nutrient cycling, the fate of C added to soil via plants and the specific effects of the microbial population (the composition of which can now be rapidly and comprehensively identified by DNA profiling). Although the incompleteness of this understanding has provided an opportunity for alternative practitioners to promote methods of soil and land management that appear attractive, we do not yet have a scientific underpinning for these methods.

Furthermore, soil scientists have failed to communicate effectively with the public, the media and policymakers to gain recognition for their achievements and to encourage the investment necessary for the prosecution of evidence-based research. Soil science needs communication champions with credible stories to tell.

**Author Contributions:** Conceptualization, R.E.W; Data Curation R.E.W., M.P., Writing-Original Draft Preparation, R.E.W.; Writing-Review & Editing, R.E.W., M.P.

**Funding:** This research received no external funding.

**Acknowledgments:** We acknowledge valuable comments from A.B. McBratney of the University of Sydney, R.G.V. Bramley of CSIRO, Adelaide and three anonymous reviewers.

**Conflicts of Interest:** The authors declare no conflict of interest.

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
