# Peer review of "Orthodox Soil Science versus Alternative Philosophies: A Clash of Cultures in a Modern Context"

_sustainability, doi:10.3390/su11102919_

Round 1

Reviewer 1 Report

Please find attached my fairly comprehensive reviewers report in word format.  I attach this word document because it contains bold font, italics font and yellow highlight to assist the authors to improve their manuscript.

Author Response

Response to reviewer 1

We thank this reviewer for his constructive comments and we have accepted his suggestions. In particular:

1. Using the term ‘orthodox soil scientists dealing with land management’) in l.10 and elsewhere.

2. L. 111 Adding ‘research capabilities and achievements of CSIRO’.

3. Ll. 177–187. Taking up his suggestion of a revised paragraph giving two examples of how scientists and the farming community (citizen scientists) have collaborated on catchment-related projects (with references). We have also expanded on the SGS program at this point, as suggested by reviewer 2, to give another example of scientists collaborating productively with farmers in field research.

Reviewer 2 Report

Review Soil Systems 457421

This is a well written, important paper that clearly addresses one of the 5C’s: “connectivity”. I suggest the authors refer to this particular C when revising their manuscript.

The authors describe a phenomenon that is widespread and does not receive sufficient attention. The presented examples convincingly demonstrate the fact that statements are made that are not supported by robust data. Attending the Soil Security meeting, I appreciated the fact that farmers could present their stories. This is new for scientific meetings. Several of them had well documented case studies but some also seemed to be particularly focused on illustrating and enjoying their own excellence.

The paper thus  illustrates the gap between science and (parts of) society. But what to do about it? One can only agree that “communication, engagement, confidence and credibility” should receive more attention ( again, I suggest that these aspects are all covered by the “connectivity” issue.). But when the authors discuss “communication and credibility” , they again review two studies on P and moisture that clearly produce nonsense. Even though absolutely true, this will not generate “engagement” with the other parties let alone “credibility”. This is only human: telling people that they are dead-wrong leads to a negative, defensive reaction and lack of meaningful communication, engagament , confidence and credibility.

I would suggest that it would be good to describe the apparently successful  SDG study in much more detail. Why was this succesfull in terms of “connectivity”? This could show the way to the future. There are other examples in the US, where “translational” research with more interaction with farmners receives increasing emphasis. So, the authors could suggest that successful joint studies are the best recipe to achieve “connectivity”. But involving serious research practices also implies that “nonsense”will be revealed. This will have to be accepted.

I certainly agree that economic aspects should be part of the picture. We did a farmer-initiated study on land use where a Life Cycle Analysis was performed including an economic analysis and that added essential information.

Dolman, M.A.; Sonneveld, M.P.W.; Mollenhorst, H.; de Boer, I.J.M. 2014. Benchmarking the economic, environmental and societal performance of Dutch dairy farms aiming at internal cycling of nutrients.

J. Clean. Prod.  73, 245–252.

At the same time, I believe that some aspects could in all fairness be mentioned as well, showing that the scientific arena is not all filled with roses. For example: (1) the “green”revolution has been a triumph of plant-breeders but has been an environmental disaster in many areas due to excessive fertilizatiion and use of pesticides. That is, of course, being corrected now but we can’t deny that industrial agriculture has left its negative traces. (2) soil science has had its own subcultures: glas-bead soil physics, ignoring preferential flows; soil chemistry focusing on grounded out-of-context soil samples, little emphasis on soil biology for a long time. Unfortunate to see now that soil biologists fall in the same trap by too often ignoring physical and chemical soil conditions while doing their studies.

The authors have to decide how far they will go in describing background aspects in this study but recognizing one’s own limits can help to close the gap with critical outsiders.

Be that as it may, I believe that we have to defend the scientific standards and this could be stated clearly, also I this paper. Formulating a hypothesis, testing with methods that produce reproducible results, applying statistics when formulating conclusions. And then: falsifying. This is needed to obtain data that can be extrapolated: a good point made by the authors.

Ideology ( and that’s what the mentioned books boil down to) and science don’t go together! Period! But people are receptive listening to stories of a past that never existed and we have to be aware of that. Indeed, why are there seem to be few  books with a positive focus on soil- and agricultural research?

I have argued with sociologist colleagues who stated that all soil scientists had to do was to listen to farmers and start from there. That degrades the science to a docile service and we know that it can do much more by showing new ways that no farmer  could ever imagine.

Author Response

Response to reviewer 2

1. Ll. 141-142. We have indicated that our contribution can fall under the heading a ‘connectivity’, one of the 5C’s identified previously under ‘dimensions of soil security’.

2. As indicated above, in ll. 177–181, we have expanded on the workings of the SGS program. Our reference (4) gives a very complete analysis of the achievements of this program and the reactions of the participants.

3. We imply the importance of cost-benefit analysis in the reporting of alternative practice systems in l. 47.

4. We have specifically confined our comments to land management issues in Australia. So as not to dilute our message about the clash of cultures in Australia, we do not want to discuss the pros and cons of the ‘green revolution’ or of narrowly focused subcultures that may exist in soil science globally. However, we have added an example, with a reference, of the shortcomings of narrowly focused research in an Australian context in ll. 103–106.

5. One of the main aims of this paper is to advocate that soil scientists need to engage constructively with alternative practitioners, but at the same time maintain a rigorous approach to the investigation of natural phenomena.

Reviewer 3 Report

This paper is unremarkable in its conclusions - everyone can agree that soil scientists should communicate more effectively with the public. However, I criticise the paper mainly for its tone - at the start, especially, it sounds defensive of the profession of soil science as against broad-brush proponents of regenerative farming. Perhaps there is some justification in the paragraph on l. 48-57 where the two main Australian authors on regenerative agriculture are quoted as saying that scientists cannot provide answers to the obvious questions posed by landscape deterioration, and such effects as erosion, particularly under stresses imposed by increasingly common droughts. It might have been less confrontational for the authors to accept that there are problems and to accept that some farmers have apparently had successes in improving their situations. Yes, their results are anecdotal - athough there are quantitative improvements in stock yield given in Massy's book. The farmer authors do not have the tools to describe exactly why positive changes occur - this is indeed the role of scientists. In searching for causes at the micro-scale Massy gets it wrong in at least one instance in writing about organic matter as humic and fulvic acids. Now we know better.

If the farmers can be criticised for over-emphasising the "mechanical mind" and its effects, scientists can be criticised, at least in the past for over-emphasising reductionist approaches. At least the early papers cited in l. 98-102 tend to have reductionist approaches.

The authors point to initiatives in the US and also Australia which are studying whole systems and this is good news. Maybe the problems and their urgency require some recognition of the possible validity of some regenerative agriculture approaches. Then soil scientists can help to determine what, if anything, makes them different at a more micro-level. However, there should also be a recognition that soils are not closed entities and that training in soil science may not be sufficient to understand whole systems at the micro level. There will be a need for the likes of hydrologists and ecologists to help here. A fault may lie in the separation of soil scientists from other landscape scientists and even in the tendency for discipline walls within soil science. A further problem may lie in the way that science is funded, especially in state departments of agriculture and the CSIRO. It was good to hear that the SGS program is a 6-year long program. Funding is often obtainable only from industry sources (including Meat and Livestock Australia and also e.g. Grains Research and Development Corporation). These have their particular stakeholders and may be reluctant to fund whole system long-term approaches. The same should not apply to Universities but they also have strong commitments to industry funding.

It has to be recognised that farmers, with their livelihoods bound to the land, can make useful contributions to progress, even if they are not explained at the micro-level. An example is the discovery some 60 years ago by South Australia farmer Clem Obst that addition of clay to the surface of water repellent soils overcame the problem. The reasons for this took about 30 years to sort out by spoil scientists and now technologies have been developed to improve clay addition, e.g. by delving from lower horizons. The regenerative agriculturalists should also be seen as possibly providing solutions to undoubted problems in Australian agriculture. Then scientists, in soil and related disciplines, can examine their systems and come up with formulae that best enable long-term improvements over wide areas and soil types.

Author Response

Response to reviewer 3

1. We have tried to present a balanced view of the clash of cultures in Australian land management. We have indicated that farmers frequently have demonstrated the success of their methods (l. 175) and that scientists have been ready to engage with them (l. 176). We have expanded on examples where this engagement has been successful (ll. 177–185). We have also acknowledged that in some cases ‘reductionist’ approaches to problem-solving have shown their limitations (ll. 103–106). However, we point out that much land management research has now evolved from simple plot experiments to catchment scale studies (ll. 106–109).

2. We acknowledge that there may validity in some of the regenerative agriculture methods. We advocate that scientists should be involved in trying to explain how such systems work to identify the situations in which such methods can be applied, given particular soil types, landscape dynamics and climates (ll. 194–203).

3. We did not think it relevant to this paper to delve into ‘training in soil science’. However, as indicated above, research into land management problems has evolved to be more holistic in approach (ll. 106–109) than the reductionist methods of the past, which suggests that education in soil science in universities has evolved too.

4. In our Conclusions we acknowledge that although scientists have made much progress over the years there is still more to be done, particularly in respect of the biology of soils in farming systems. We advocate (ll. 188–189) that this can be done when alternative practitioners work with scientists, rather than when some of the spokespeople for the former dismiss the potential contributions that science can make.

Round 2

Reviewer 3 Report

Thisw manuscript has dealt with my earlier reservations and is now OK

Author Response

The reviewer says that he/she is now satisfied with the amended paper.